# Is there an association between long-term antibiotics for acne and subsequent infection sequelae and antimicrobial resistance? A systematic review protocol

Ketaki Bhate [1], Liang-Yu Lin [1], John Barbieri,[2] Clémence Leyrat,[3] Susan Hopkins,[4] Richard Stabler,[5] Laura Shallcross,[6] Liam Smeeth,[1] Nick A Francis,[7] Rohini Mathur [1], Sinéad M Langan,[1] Sarah-Jo Sinnott[1]

SML and S-JS are joint senior authors.

For numbered affiliations see end of article.

**Correspondence to**
Dr Ketaki Bhate;
ketaki.bhate@lshtm.ac.uk

## ABSTRACT

**Introduction** Antimicrobial resistance (AMR) is a global health emergency. Acne vulgaris is a highly prevalent condition and the dominant role antibiotics play in its treatment is a major concern. Antibiotics are widely used in the treatment of acne predominantly for their anti-inflammatory effect, hence their use in acne may not be optimal. Tetracyclines and macrolides are the two most common oral antibiotic classes prescribed, and their average use can extend from a few months to several years of intermittent or continuous use. The overall aim of this systematic review is to elucidate what is known about oral antibiotics for acne contributing to antibiotic treatment failure and AMR.

**Methods and analysis** A systematic review will be conducted to address the question: What is the existing evidence that long-term oral antibiotics used to treat acne in those over 8 years of age contribute towards antibiotic treatment failure or other outcomes suggestive of the impact of AMR? We will search the following databases: Embase, MEDLINE, the Cochrane Library and Web of Science. Search terms will be developed in collaboration with a librarian by identifying keywords from relevant articles and by undertaking pilot searches. Randomised controlled trials, cohort and case-controlled studies conducted in any healthcare setting and published in any language will be included. The searches will be re-run prior to final analyses to capture the recent literature. The Cochrane tool for bias assessment in randomised trials and ROBINS-I for the assessment of bias in non-randomised studies will be used to assess the risk of bias of included studies. GRADE will be used to make an overall assessment of the quality of evidence. A meta-analysis will be undertaken of the outcome measures if the individual studies are sufficiently homogeneous. If a meta-analysis is not possible, a qualitative assessment will be presented as a narrative review.

**Ethics and dissemination** Ethical approval is not required for this systematic-review. The results will be published in a peer-reviewed journal and any deviations from the protocol will be clearly documented in the published manuscript of the full systematic-review.

**PROSPERO registration number** CRD42019121738.

## Strengths and limitations of this study

► To our knowledge, this is the first comprehensive systematic review that will address the use of oral antibiotics for acne and their contribution to antimicrobial resistance.

► Screening, data extraction and quality assessment will be undertaken independently by three medically qualified researchers with training in systematic review methodology, thereby ensuring scientific rigour, transparency and repeatability.

► There are no date or language restrictions; however, this systematic review will not examine the grey literature.

## INTRODUCTION

The future effectiveness of antibiotics is in jeopardy with the WHO declaring the threat of antimicrobial resistance (AMR) as a most urgent crisis.[1] Future deaths from infections as a result of AMR without any intervention is estimated at 10 million per year and by 2050 the cost of AMR could reach 100 trillion US Dollars.[2]

Topical and oral antibiotics are commonly prescribed for the treatment of acne vulgaris, a chronic skin disorder with onset predominantly in adolescence. Given the psychosocial consequences and the potential for permanent disfigurement with scarring, it is imperative that people with acne receive effective treatment.[3 4] Prevalence studies show that 80% to 100% of teenagers have acne and that 20% are moderately-to-severely affected. The high prevalence means that both topical and oral antibiotics are used in a large proportion of the adolescent population and for variable durations ranging from 6 weeks to many months, and in some cases, several years.[5 6] Differences between international guidelines regarding duration of treatment is one of the

reasons that antibiotics for acne are used for significantly longer than recommended as there is uncertainty about the optimal duration of treatment.[6–11] Tetracyclines and macrolides are the two of the most common oral antibiotic classes prescribed for acne with varying durations of average use depending on treatment setting and between different countries.[6 12]

The overuse of antibiotics is a known cause of AMR as repeated and sustained exposure allows microbes to develop mechanisms to avoid the effects of the drugs designed to treat them and allows selection in favour of bystander or commensal bacteria with resistance subsequently causing invasive infection. Acne is aetiologically multifactorial and we already know that some strains of *Cutibacterium acnes* (formally *Propionibacterium acnes*), the bacteria pathophysiologically associated with acne, are now resistant to commonly used antibiotics in acne, making their initial use as anti-microbial agents futile.[13 14] However, we do not know how these long-term antibiotics for acne may attenuate microbiota elsewhere at other body sites and the ability of other bacteria at other infective sites to withstand the effect of antibiotics. Despite this, the anti-inflammatory effect and proven efficacy of antibiotics in treating acne ensures their continued use,[15] although their effects may not be sustained. Considering the relationship between long-term exposure to antibiotics and AMR, this practice may not be optimal.

The effects long-term antibiotics for acne have on future infections caused by resistant organisms, subsequent antibiotic treatment failure or the rate of infections (or any other measures which may indicate antimicrobial resistance) and how long any effect may last, is not yet known and has not been systematically reviewed in the literature before. While antibiotic stewardship programmes have been shown to be effective[16] in other settings, to ensure their successful execution, robust evidence must be generated to show that using antibiotics in the treatment of acne has important implications for future infective episodes and resistance sequelae. Until there is evidence of how the use of oral antibiotics for acne may cause AMR, changing current practice will be challenging.[17]

Given the global health emergency of AMR and the dominant role antibiotics play in the treatment of acne—a highly prevalent and ubiquitous skin condition worldwide, there is a clearly defined evidence gap which needs to be urgently addressed.[18] This systematic review aims to establish what is already known about resistance sequelae for those with acne who are treated with long-term topical or oral antibiotics.[19]

## METHODS AND ANALYSIS
### Literature search strategy
We will search the following databases; Embase, MEDLINE, the Cochrane Library and Web of Science. We will develop the search terms by identifying keywords from relevant articles and by undertaking pilot searches to identify index or MedicalSubject Heading (MeSH)

terms. We will modify the search terms according to each database, for example, the MeSH terms in MEDLINE and Emtree terms in Embase. Searches will be undertaken by the lead author who has medical and search training in collaboration with a librarian. Search strategies will be reviewed by all authors. The searches will be kept as broad as possible, for example, by using the 'explode' function on the Ovid platform to maximise the number of relevant articles. The search strategy is available to view in the accompanying supplement (online supplementary file 1). Searches were undertaken on the 19th of July 2019 and date back to inception of the databases.

### Eligibility criteria
#### Inclusion criteria
To address the question, the following inclusion criteria will apply:
- ► A study population including participants aged over the age of 8 in any healthcare setting with acne vulgaris.
- ► Original studies will be eligible for assessment for inclusion if they address the specific research question.
- ► Randomised controlled trials (of any trial design).
- ► Observational studies limited to cohort and case-control studies.
- ► We will include conference abstracts if the full paper is unpublished and can be obtained from the authors.

#### Exclusion criteria
- ► Ecological studies and studies that do not assess temporality such as case-series and case reports.
- ► We will exclude unpublished studies, ongoing studies and the grey literature.
- ► In addition, studies which only look at antimicrobial resistance in *Propionibacterium acnes* or *Cutibacterium acnes*.
- ► Studies including people who are under the age of 8 exclusively will be excluded. The age of 8 was chosen as acne vulgaris is unlikely to present in younger children and in addition, tetracyclines are not recommended in younger children—the British National Formulary recommends tetracyclines are given to children aged 12 years and above.
- ► Studies including people who are treated with antibiotics for other acne subtypes, for example, hidradenitis suppurativa or drug-induced acne.

### Exposure
At least 28 days of continuous (daily doses) oral antibiotics for acne vulgaris, the duration helping to ensure treatment is not targeted at an acute infective episode and, in addition, 28 days is the minimum duration a prescription will be issued for an antibiotic treatment of acne. The exposure is likely to include commonly used antibiotic classes—tetracyclines, macrolides and dihydrofolate reductase inhibitors, however there will be no limits placed on the antibiotic class used to treat acne. We have excluded the use of topical antibiotics are these are

less likely to have an effect at sites other than the skin to where they are applied.

## Comparator

No exposure to long-term oral antibiotics within an acne population or within a general population.

## Outcome

The primary outcome is antibiotic treatment failure or any infection caused by a resistant organism. The secondary outcome is the detection of resistant organisms without a clinical infection, rate of infection or changes to the microbiota profile, for example, with the colonisation of resistant microbiota without a clinical infection or different microbiota in a sampled site compared with baseline prior to having received a long-term antibiotic for acne. Any measure (including proxy measures) will be included, for example, laboratory measures (such as an elevated C-reactive protein or positive culture in the case of an infection at any body site), patient observations (such as an elevated temperature and/or pulse rate which may indicate an infective process) or proxy measures that may have been used in epidemiological studies, for example, difficult to treat infections which may indicate a resistant infection. Each outcome will be assessed separately. The outcome can occur at any time point after at least 28 days of continuous oral antibiotic exposure for acne; we will stratify according to the length of follow-up, for example, up to 6 months, 6 months to 1 year, 1 year to 2 years, and so on.

## Potential confounding variables/effect modifiers

Confounding factors that may be considered by studies investigating treatment failure or AMR as a result of long-term antibiotics for acne are: age, sex, socioeconomic status, treatment adherence, medical conditions such as primary immunodeficiency, diabetes, asthma, cancer requiring immunosuppressive medication, recent hospitalisation within the last 6 months, repeated admissions to hospital, any recurrent infections, other prescribed medication in particular immunosuppressive therapy including oral corticosteroids, smoking, alcohol use and ethnicity. We will also explore effect modification. The inclusion of these confounding factors will be acknowledged in the bias assessment of each study along with a statement of the direction and magnitude of bias their omission may be associated with.

## Eligibility assessment and data extraction
### Phase 1

Covidence, an online literature review data management programme, will be used to facilitate the systematic review process inclusive of title and abstract screening, full paper retrieval and storage and decisions on which papers to include at full text review. In the first phase, all titles and abstracts will be uploaded to Covidence. Duplicates will then be removed by the lead reviewer (KB). Three reviewers (KB, L-YL and JB) will then independently screen the search results based on title and abstract. Each

title/abstract will require two votes. Consensus will be achieved on the number of titles and abstracts to include in the full study review. Any disputes will be resolved by the involvement of a fourth reviewer (SML).

### Phase 2

Full text papers will be assessed independently by the reviewer pairs using a standardised data extraction form. The extraction tool will be piloted using the first three included records, after which modifications may be made following discussion with other members of the review team. The quality of the studies will be scored using assessment tools and free text explanations for the score given will be included on the score sheet. Any disagreements will be discussed by the three reviewers (KB, L-YL and JB) and in instances of disagreement, a fourth reviewer (SML) will make a final decision. If ambiguity still remains after the full text is obtained, the study authors will be contacted for further clarification.

## Data items

Three data domains will be extracted:

### Data relating to study design

Author, country, specific study design, the year the study was conducted or the years over which the data were collected. Healthcare setting, the number of study participants, the ages of the participants and the gender balance will be collected for the whole population under study, including the comparator group. If the study is a trial, then specifics of the study design such as randomisation, allocation concealment and blinding will be noted.

### Data relating to exposure

The dose, frequency and antibiotic used, the median/mean length of treatment of acne with the antibiotic, the definition of long-term treatment with antibiotics used in the study, the number of participants exposed to antibiotics and if multiple courses are prescribed, the length of time between antibiotic courses and the intervention applied to comparators.

### Data relating to outcomes

The measure of antibiotic treatment failure or AMR and the degree of antibiotic treatment failure or AMR, for example, repeat course required, hospitalisation or death. The length of follow-up will be stratified.

## Study quality assessment

Each study will be critically appraised by reviewers. The Cochrane tool for bias assessment in randomised studies and the ROBINS-I tool for the assessment of bias in non-randomised studies will be used to assess the risk of bias in included studies.[20–22] GRADE (Grading of Recommendations Assessment, Development and Evaluation) will be used to make an overall assessment of the quality of evidence.[22] Pairs of reviewers will make independent assessments of the risk of bias. Markers of bias depending on study design included in the aforementioned scoring

tools will include factors such as the method of participant selection, follow-up, randomisation, adjustment for confounding and measurement error of exposures or outcomes. If a proportion of studies have a high risk of bias found using the scoring tool we will do a sensitivity analysis excluding them.

## Data synthesis/statistical analysis

We will analyse interventional and observational studies separately. If there is homogeneity across studies and a meta-analysis is possible, we will generate a pooled effect estimate for those exposed to long-term antibiotics and those unexposed within each category of study design. If there are a sufficient number of studies, subgroup analyses will be undertaken, for example, by class of antibiotic and antibiotic treatment duration. The $I^2$ statistic will be used to assess heterogeneity.[23] Sources of heterogeneity may include methodology, age of participants, study duration, the confounding factors considered, the exposure (ie, length/duration, the class of antibiotic), the comparators and the outcomes measured. If heterogeneity is above 50% we will not undertake a meta-analysis. If studies are sufficiently homogeneous with regard to exposures, comparators and outcomes, a random effects model will be used to generate a pooled relative risk and its 95% CI. Study characteristics and the effect estimate for the association between antibiotics for acne and the specific measure of AMR will be clearly presented. We will also do a sensitivity analysis using a fixed effects model. Publication bias will be assessed using Funnel plots and Egger tests.[24] Forest plots will be presented. All statistical analyses will be performed using Stata. If quantitative synthesis is not possible due to heterogeneity, we will conduct a narrative synthesis. We will also study each category of outcome measure separately: for example, laboratory-based measures of resistance or outcome measures thought to be proxies for AMR using routinely collected health records. Given the breadth of outlined outcomes, it is likely that the evidence obtained will be diverse. An overall description of the strength of the body of evidence generated using GRADE will be described.[21]

The study will be reported following Preferred Reporting Items for Systematic Reviews and Meta-Analyses guidance.[19]

## Patient and public involvement

This systematic review has been informed by the results of the Acne Priority Setting Partnership (PSP) (acnepsp. org) in collaboration with the James Lind Alliance (www. jla.nihr.ac.uk). Over 6000 responses were collated and voted on to give a top 10 list of treatment uncertainties. Two of these top 10 uncertainties will be addressed with this systematic review:

1. What is the correct way to use antibiotics in acne to achieve the best outcomes with the least risk?
2. What management strategy should be adopted for the treatment of acne in order to optimise short-term and long-term outcomes?

In addition, five people comprising members of the public and patients with acne or their carers will attend a focus group to help write the summary which will be used to disseminate the results of this systematic review to the public.

## Ethics and dissemination

As this is a systematic review, ethical approval was not required. This systematic review protocol was registered on the 8th of April 2019 on the International Prospective Register of Systematic Reviews (PROSPERO). Any amendments to the protocol will be updated and published on the PROSPERO website with clear notes of where specific changes were made with detailed explanations of why. The results of this systematic review will be submitted for peer-review publication.

**Author affiliations**
[1]Department of Non-communicable Disease Epidemiology, London School of Hygiene and Tropical Medicine Faculty of Epidemiology and Population Health, London, UK
[2]Perelman School of Medicine, Department of Dermatology, University of Pennsylvania, Philadelphia, Pennsylvania, USA
[3]Department of Medical Statistics, London School of Hygiene and Tropical Medicine, London, UK
[4]Public Health England, London, UK
[5]London School of Hygiene and Tropical Medicine Faculty of Infectious and Tropical Diseases, London, UK
[6]Institute of Health Informatics, Faculty of Pop Health Sciences, University College London, London, UK
[7]School of Primary Care, Population Sciences and Medical Education, University of Southampton, Southampton, UK

**Contributors** KB wrote the protocol and led the study. S-JS, SML and RM supervised the writing process and contributed equally. L-YL and JB contributed to the screening and review process. CL, RS, LS, SH, NAF and LS form an advisory group and reviewed the protocol.

**Funding** KB is funded by an NIHR Doctoral Research Fellowship DRF-2018-11-ST2-066. Professor Sinéad Langan is funded by a Wellcome Trust Senior Clinical Fellowship (205039/Z/16/Z). JB is supported by the National Institute of Arthritis and Musculoskeletal and Skin Diseases of the National Institutes of Health under award number T32-AR-007465. LS is funded by an NIHR Clinician Scientist Award CS-2016-16-007.

**Disclaimer** The views expressed are those of the author(s) and not necessarily those of the NHS, the NIHR or the Department of Health and Social Care.

**Competing interests** JB is supported by the National Institute of Arthritis and Musculoskeletal and Skin Diseases of the National Institutes of Health under award number T32-AR-007465 and receives partial salary support through a Pfizer Fellowship grant to the Trustees of the University of Pennsylvania.

**Patient consent for publication** Not required.

**Provenance and peer review** Not commissioned; externally peer reviewed.

**ORCID iDs**
Ketaki Bhate http://orcid.org/0000-0001-5509-4428
Liang-Yu Lin http://orcid.org/0000-0003-4720-6738
Rohini Mathur http://orcid.org/0000-0002-3817-8790

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
