## [Reviewer comments · BMJ Open]

ARTICLE DETAILS

TITLE (PROVISIONAL)	Is there an association between long-term antibiotics for acne and subsequent infection sequelae and antimicrobial resistance? Systematic review protocol
AUTHORS	Bhate, Ketaki; Lin, Liang-Yu; Barbieri, John; Leyrat, Clémence; Hopkins, Susan; Stabler, Richard; Shallcross, Laura; Smeeth, Liam; Francis, Nick A.; Mathur, Rohini; Langan, Sinead; Sinnott, Sarah-Jo

VERSION 1 – REVIEW

REVIEWER	Hei Sung Kim The Catholic University of Korea, Seoul, Korea
REVIEW RETURNED	13-Nov-2019

GENERAL COMMENTS	Antimicrobial resistance from the use of antibiotics in Acne is of great interest and importance as it is universally used as the authors mention. 1. In page 7, line 15, it is stated that acne is a chronic, non-infectious skin disorder, but since we are in the process of understanding the role of microbiota in acne, I think it would be better to just mention that acne is a chronic inflammatory disorder. It applies to same to page 7, line 40. I wouldn't want to jump to conclusions that acne is not an infectious disease.2. As for the outcome (page 9, line 4), I wonder if we could attribute treatment failure to "only" AMR from acne treatment. That is, in clinical practice, there are patients you have been exposed to long-term antibiotics from other diseases, and some have too much of an oily skin which is not sufficiently controlled by oral antibiotics. How would you define "treatment failure"? would these be patients who initially responded well to treatment and later on did not, or those who showed little response to treatment from the beginning?3. How are we to know if there is an infection caused by a resistant organism? Are we including studies which exclusively analyze the functional characteristics of the microbiota?4. Since acne is mostly localized on the face, even if there is secondary infection I would be surprised to observe significant elevation in C-reactive protein or increases in temp/heart rate (page 10). I am not sure what the secondary outcome is aiming at.
---

REVIEWER	Steven Feldman Wake Forest Baptist Health
	I have received research, speaking and/or consulting support from a variety of companies including Galderma, GSK/Stiefel, Ammiral, Alvotech, Leo Pharma, BMS, Boehringer Ingelheim, Mylan,

	Celgene, Pfizer, Ortho Dermatology, Abbvie, Samsung, Janssen, Lilly, Menlo, Merck, Novartis, Regeneron, Sanofi, Novan, Qurient, National Biological Corporation, Caremark, Advance Medical, Sun Pharma, Suncare Research, Informa, UpToDate and National Psoriasis Foundation. I also consult for others through Guidepoint Global, Gerson Lehrman and other consulting organizations. I am founder and majority owner of www.DrScore.com. I am a founder and part owner of Causa Research, a company dedicated to enhancing patients' adherence to treatment.
REVIEW RETURNED	06-Jan-2020

GENERAL COMMENTS	I was excited by the idea that there were be a good protocol for a good study to evaluate long-term effects of antibiotics on antimicrobial resistance but was disappointed when I saw that this only plans to review existing studies. Ah, well. For a systematic review, this protocol looks fine. I'm not excited by it, and I'm not we need to have a protocol published for a systematic review. But it doesn't hurt to publish it.
--

REVIEWER	Dr. Helen Stagg University of Edinburgh, UK
	We currently have a review in progress on the length of macrolide use and antimicrobial resistance. For the protocol, see Divala et al. Systematic Reviews (2018) 7:246
REVIEW RETURNED	03-Feb-2020

GENERAL COMMENTS	Thank you for the opportunity to review this systematic review protocol on the use of antibiotics for acne and the association with antimicrobial resistance. I note here that my expertise is in systematic reviews, rather than the specific condition and antibiotics under study. This is a good and interesting protocol; my comments solely relate to clarifying its contents to aid the reader and querying the breadth of the work. MAJOR General a) Have you checked your submission against the PRISMA-P checklist? I cannot see if you have included this with your submission. b) I would add in a limitation on the likely diverse evidence that you will obtain, given the breadth of your outcomes, etc. Methods c) Why have you chosen to include oral, but exclude topical, antibiotics from your analysis? Please justify. d) Line 39, page 5- you have stated that searches will be undertaken in July 2019. How far through this review are you, currently? Given this point about dates, please alter the preceding text to the past tense. e) Apologies if I am missing something, but I could not find the supplement with the search terms in the submission. f) Line 3, page 8- the first sentence of this paragraph seems to have gone a little funny during the editing process. By 'antibiotic treatment failure' do you mean the failure of the subsequent antibiotic treatment of any other condition working to cure that condition? How will you consider a mismatch between the original and subsequent treating antibiotics i.e. whether the original antibiotic could induce relevant drug resistance for the later
---

antibiotic? Ditto for 'infection caused by a resistant infection'- does this have to be resistance towards the antibiotic originally used to treat the infection? How far into the future should a new infection be before it cannot be related to the original treatment (as, presumably, bacteria in question need to have been in place at the time of initial treatment)?

g) Please clearly designate what you mean by 'infection' and 'without an infection'.

h) By 'changes to flora' do you mean in terms of the bacteria, etc. present or in terms of the gain of specific drug resistance?

i) How will you calculate rate of infection and how does this relate to your research questions?

j) It would be helpful to list your different study outcomes in a table and explain them more fully e.g. why they are related to your study question/antimicrobial resistance, providing references and the measures/proxy measures of each that you are likely to encounter (I see that some of this is in your later 'outcomes' section).

k) Page 8, line 31- do you mean 'in', rather than 'to'? Please state clearly that this criterion is because you are interested specifically in subsequent resistance in bacteria causing other disease(s).

l) Page 8, line 34- why not place the age cut-off at 12 years?

m) Page 8, line 41- you say 'other subtypes', but do not state which ones you are interested in during the inclusion criteria.

n) Page 8-9- please merge these ICO sections with the ones above, to aid the reader and avoid repetition.

o) Page 9, line 28- which of these are potential confounders and which potential effect modifiers?

p) Page 9, line 31- how about treatment adherence as a confounder?

q) Page 10, line 34- which other characteristics of the study population will you collect e.g. the type of control population? Make sure that data on all your potential confounders/effect modifiers are captured too.

r) Page 10, line 35- you will also need to capture the dose of medication and frequency with which it was administered. It is better to document the precise antibiotic rather than the class at the extraction stage; you can always group things together later.

s) Page 11, line 17- stepping back from the I2 statistic, given the huge array of outcomes (disease at different sites with different bacteria) and exposures you propose to include, do you think that a meta-analysis will be possible across the entire dataset? Why not narrow down your review to produce a more homogenous dataset? Why risks and not rates (how will you extract these different types of data)?

MINOR
Abstract

t) Line 11, page 4- anti-inflammatory effect, as opposed to what?

u) Line 18, page 4 (aim) is very broad and contrasts sharply to the specificity of line 25's research question. How about condensing these two things together?

v) Line 27, page 4- define 'long term'.

w) Line 28, page 4- what is meant by 'increased infectious outcomes'?

x) Line 30, page 4- roughly what areas/topics do you expect the search terms to cover?

y) Line 42, page 4- references are not necessary in the abstract.

z) Line 42, page 4- by 'quantitative assessment', do you mean some type of meta-analysis?

	Methods and analysis aa) Line 52, page 7- small point, but it would helpful to state that the population under study has acne. bb) Page 9, line 55- how many reviewers will read each abstract?
--	--

VERSION 1 – AUTHOR RESPONSE

Reviewer 1

1. *In page 7, line 15, it is stated that acne is a chronic, non-infectious skin disorder, but since we are in the process of understanding the role of microbiota in acne, I think it would be better to just mention that acne is a chronic inflammatory disorder. It applies to same to page 7, line 40. I wouldn't want to jump to conclusions that acne is not an infectious disease.*
 - i. There is much debate surrounding acne as an infectious disease. We have removed the words non-infectious from both page 7 line 15 and page 7 line 40.
2. *As for the outcome (page 9, line 4), I wonder if we could attribute treatment failure to "only" AMR from acne treatment. That is, in clinical practice, there are patients you have been exposed to long-term antibiotics from other diseases, and some have too much of an oily skin which is not sufficiently controlled by oral antibiotics. How would you define "treatment failure"? would these be patients who initially responded well to treatment and later on did not, or those who showed little response to treatment from the beginning?*
 - i. We are not looking for antibiotic treatment failure with acne as an outcome, but are more focused on any infectious diseases occurring subsequent to having had antibiotics for acne. We have changed the outcome section to make this clearer to the reader.
3. *How are we to know if there is an infection caused by a resistant organism? Are we including studies which exclusively analyse the functional characteristics of the microbiota?*
 - i. We would know if an infection is caused by a resistant organism if a study undertook laboratory antibiotic resistance testing for an isolated organism. One of our secondary outcomes is rates of infection – here there will be some uncertainty about the extent to which individual infection could be attributed to previous antibiotic use, but in that case, the important outcome would be evidence of higher rates of infection.
 - ii. Regarding changes to the microbiota, we are looking for studies which assess changes to microbiota as a result of long-term antibiotics for acne as well as studies assessing microbiota resistant to antibiotics.
4. *Since acne is mostly localized on the face, even if there is secondary infection I would be surprised to observe significant elevation in C-reactive protein or increases in temp/heart rate (page 10). I am not sure what the secondary outcome is aiming at.*
 - i. We are interested in any resistant infection (primary outcome) or any infection at any body site (secondary outcome) in those who have treated with long-term oral antibiotics for acne. We have clarified this within the body of the text.

Reviewer 2

- 1 *I was excited by the idea that there were be a good protocol for a good study to evaluate long-term effects of antibiotics on antimicrobial resistance but was disappointed when I saw that this only plans to review existing studies. Ah, well. For a systematic review, this protocol looks fine. I'm not excited by it, and I'm not we need to have a protocol published for a systematic review. But it doesn't hurt to publish it.*
 - i. Reviewing the existing evidence on the effect on long-term oral antibiotics on antibiotic treatment failure and antimicrobial resistance forms the first step of a larger project. Reviewing the evidence will allow us to gain an understanding of what we know already and familiarise us with the strengths and weaknesses of the current literature. We will then use the knowledge from the review to design a study to

address the knowledge gap and weaknesses uncovered. I look forward to sharing the outcome of the completed systematic review.

Reviewer 3

MAJOR

- A. *Have you checked your submission against the PRISMA-P checklist? I cannot see if you have included this with your submission.*
- i. Apologies this was not included in the original submission. It was completed and has been uploaded now.
- B. *I would add in a limitation on the likely diverse evidence that you will obtain, given the breadth of your outcomes, etc.*
- i. We have added a line stating this into the latter part of the analysis section (Page 11, line 29).
- C. *Why have you chosen to include oral, but exclude topical, antibiotics from your analysis? Please justify.*
- i. Oral antibiotics will have systemic effects and therefore may cause changes to microbiota and subsequent resistant infections elsewhere at other body sites. We have added a line to clarify the reason for excluding topical antibiotics in the exposure section (page 8, line 30).
- D. *Line 39, page 5- you have stated that searches will be undertaken in July 2019. How far through this review are you, currently? Given this point about dates, please alter the preceding text to the past tense.*
- i. The searches were completed in July 2019 and the review is now near completion. We are evaluating evidence with GRADE and writing the manuscript. The tense has been modified from 'will be' to 'were'.
- E. *Apologies if I am missing something, but I could not find the supplement with the search terms in the submission.*
- i. This has been uploaded now.
- F. *Line 3, page 8- the first sentence of this paragraph seems to have gone a little funny during the editing process. By 'antibiotic treatment failure' do you mean the failure of the subsequent antibiotic treatment of any other condition working to cure that condition? How will you consider a mismatch between the original and subsequent treating antibiotics i.e. whether the original antibiotic could induce relevant drug resistance for the later antibiotic? Ditto for 'infection caused by a resistant infection'- does this have to be resistance towards the antibiotic originally used to treat the infection? How far into the future should a new infection be before it cannot be related to the original treatment (as, presumably, bacteria in question need to have been in place at the time of initial treatment)?*
- i. The editing has been corrected.
 - ii. By antibiotic treatment failure we mean the failure of subsequent antibiotic treatment when an antibiotic is used to treat any other infective condition. The treatment failure relates to those antibiotics working to treat that infection e.g. a urinary tract or lower respiratory tract infection.
 - iii. The working hypothesis is that there may be some cross resistance between the different classes of antibiotics. For example, if a macrolide is given long-term to treat acne, then we would also like to include studies which assess antibiotic treatment failure or resistance to other classes of antibiotic e.g., penicillins or tetracyclines. This might occur due to changes in microbiota which become resistant, or with regard to infection risk, depletion of microbiota which subsequently lead to an increased risk of infection. (The intestinal microbiota: Antibiotics, colonization resistance, and enteric pathogens. Kim S et al, Immunol Rev, 2017 Sep;279(1):90-105).
 - iv. We have not set restriction on the period of follow up. There is currently some evidence that microbiota are altered for up to one year and there are suggestions that the effect of antibiotics on microbiota may last longer. (Long-term impacts of antibiotic

exposure on the human intestinal microbiota, Jernberg et al, Microbiology. 2010, 156, 3216-3223).

- G. *Please clearly designate what you mean by 'infection' and 'without an infection'.*
- i. We have clarified this by adding in clinical infection (page 9, line 6) i.e. those who have resistant organisms detected but are not symptomatic clinically.
- H. *By 'changes to flora' do you mean in terms of the bacteria, etc. present or in terms of the gain of specific drug resistance?*
- i. Both. Any microbiota which have become resistant to an antibiotic compared to baseline pre-antibiotic, or just a change in microbiota profile of a sampled site from baseline before antibiotic treatment. I have clarified this in the text in the outcome section (page 9).
- I. *How will you calculate rate of infection and how does this relate to your research questions?*
- i. The number of clinical infections within in a defined time period in a population who have been treated with oral antibiotics for acne. This may relate to altered microbiota as a result of a long-term antibiotic thereby allowing resistant bacterial to colonise which may then contribute to an infection, or as studies suggest, an increased rate of infection.
- J. *It would be helpful to list your different study outcomes in a table and explain them more fully e.g. why they are related to your study question/antimicrobial resistance, providing references and the measures/proxy measures of each that you are likely to encounter (I see that some of this is in your later 'outcomes' section)*
- i. As per previous comments from another reviewer, we have expanded upon the outcomes within the text to explain them more fully.
- K. *Page 8, line 31- do you mean 'in', rather than 'to'? Please state clearly that this criterion is because you are interested specifically in subsequent resistance in bacteria causing other disease(s).*
- Apologies, we cannot see place in text where this might relate to. We have changed text to: In addition, studies which only look at antimicrobial resistance in *Propionibacterium acnes* (*P. acnes*) or *Cutibacterium acnes* (*C. acnes*).
- L. *Page 8, line 34- why not place the age cut-off at 12 years?*
- i. Tetracyclines are prescribed off license for acne to children younger than 12. We have therefore selected 8 as the youngest age.
- M. *Page 8, line 41- you say 'other subtypes', but do not state which ones you are interested in during the inclusion criteria.*
- i. This is clarified in the inclusion criteria under 'intervention'. We are interested in acne vulgaris only (page 7, line 27).
- N. *Page 8-9- please merge these ICO sections with the ones above, to aid the reader and avoid repetition.*
- i. Thank you. This has been modified within the document.
- O. *Page 9, line 28- which of these are potential confounders and which potential effect modifiers?*
- i. Socioeconomic status and the degree of severity of medication conditions may act as effect modifiers as well as confounders. We will explore for evidence of effect modification. This section has been updated to incorporate potential effect modifiers (page 9, line 18).
- P. *Page 9, line 31- how about treatment adherence as a confounder?*
- i. Agree, this would be an important confounder to include. This has been added to the potential confounder section (page 9, line 20). Thank you.
- Q. *Page 10, line 34- which other characteristics of the study population will you collect e.g. the type of control population? Make sure that data on all your potential confounders/effect modifiers are captured too.*
- i. Thank you. We would be collecting the setting, number, age, gender balance of the comparator group as well. We have added this in to the section 'Data relating to the study design' to make it clearer that this relates to the comparator group as well.

- R. Page 10, line 35- you will also need to capture the dose of medication and frequency with which it was administered. It is better to document the precise antibiotic rather than the class at the extraction stage; you can always group things together later.
- i. Agree with this, at the extraction stage it is better to gather as much information as possible and narrow down later if needed. We have modified the 'Data relating to exposure' section to reflect this.
- S. Page 11, line 17- stepping back from the I2 statistic, given the huge array of outcomes (disease at different sites with different bacteria) and exposures you propose to include, do you think that a meta-analysis will be possible across the entire dataset? Why not narrow down your review to produce a more homogenous dataset? Why risks and not rates (how will you extract these different types of data)?
- i. It will not be possible to do a meta-analysis. At the time of writing and submission of the protocol this was not known and so a plan for meta-analysis was included. If we were to narrow down the review, the sample size would likely be too small for a meta-analysis. Depending on the study outcomes, rates could also be used.

MINOR

- T. Line 11, page 4- anti-inflammatory effect, as opposed to what?
- i. Their anti-inflammatory effect in reducing the redness and pain associated with acne, over their bactericidal/bacteriostatic effect. The pathophysiology of acne is multifactorial and while Cutibacterium acnes may be involved in the formation of an acne lesion, there are several other pathological steps required eg, hyperseborrhoea and follicular plugging - therefore acne is not wholly considered an infectious disease. Acne often recurs or worsens on stopping the antibiotic in moderate to severe disease, therefore adding to the suspicion that the antibiotics might work largely by exerting their anti-inflammatory effect.
- U. Line 18, page 4 (aim) is very broad and contrasts sharply to the specificity of line 25's research question. How about condensing these two things together?
- i. We were outlining an overall aim on line 18, then being more specific with the objective in line 25. They both outline the same concept. We have modified this to read better in both the introduction section of the abstract (page 4, line 8) and the methods and analysis section.
- V. Line 27, page 4- define 'long term'.
- i. We were a little limited with the word count of the abstract here. The definition of long-term is expanded in body of the paper. It is any course of an antibiotic for acne > 28 days.
- W. Line 28, page 4- what is meant by 'increased infectious outcomes'?
- i. This has been replaced with antibiotic treatment failure to make it clearer.
- X. Line 30, page 4- roughly what areas/topics do you expect the search terms to cover?
- i. We are not entirely sure what this question relates to specifically, I would expect to the search terms to highlight any studies investigating the outcomes outlined in those who have been treated with long-term oral antibiotics for acne. This may include studies related to pharmacology, dermatology or antimicrobial resistance.
- Y. Line 42, page 4- references are not necessary in the abstract.
- i. Thank you. This has been removed.
- Z. Line 42, page 4- by 'quantitative assessment', do you mean some type of meta-analysis?
- i. Yes, this does indicate a meta-analysis. We have replaced the word quantitative with meta-analysis.
- AA. Line 52, page 7- small point, but it would helpful to state that the population under study has acne.
- i. This had been added to the relevant section – thank you.
- BB. Page 9, line 55- how many reviewers will read each abstract?

- i. Two reviewers will read each abstract. We have made this clearer within the text by adding the word 'abstract' next to title (page 10, line 3).
- ii.

I look forward to hearing about the outcome having addressed reviewer comments and modifying the protocol.

VERSION 2 – REVIEW

REVIEWER	Dr. Helen Stagg University of Edinburgh
REVIEW RETURNED	16-Mar-2020

GENERAL COMMENTS	Thank you for the opportunity to re-review this manuscript. I have a few tiny comments, but am otherwise happy for the paper to be published. I do not need to see a revised draft. Good job and good luck! MAJOR a) Please label which factors you consider to be potential effect modifiers in the text. Why have you chosen socioeconomic status (I am less convinced by this one)? What about drug dose and factors that affect drug metabolism? MINOR Methods b) In line with the PRISMA-P guidelines, please state who is the guarantor for the review. c) Draft search strategy- I suspect the strategy is slightly unnecessarily broad e.g. in terms of searching for infection or resistance terms, but leave it up to the authors to make this decision in light of their capacity to screen papers. And suspect that it is too late to comment on this! d) Previous comment F, response iii- please include this detail in your manuscript. e) Page 9, line 7- it would help to emphasise that 'antibiotic treatment failure' here is against a subsequent infection, not the acne.
---